# Radiomics and Delta-Radiomics Signatures to Predict Response and Survival in Patients with Non-Small-Cell Lung Cancer Treated with Immune Checkpoint Inhibitors

**DOI:** 10.3390/cancers15071968

**Published:** 2023-03-25

**Authors:** François Cousin, Thomas Louis, Sophie Dheur, Frank Aboubakar, Benoit Ghaye, Mariaelena Occhipinti, Wim Vos, Fabio Bottari, Astrid Paulus, Anne Sibille, Frédérique Vaillant, Bernard Duysinx, Julien Guiot, Roland Hustinx

**Affiliations:** 1Department of Nuclear Medicine and Oncological Imaging, University Hospital (CHU) of Liège, 4000 Liège, Belgium; 2Radiomics (Oncoradiomics SA), 4000 Liège, Belgium; 3Department of Radiology, University Hospital (CHU) of Liège, 4000 Liège, Belgium; 4Department of Pulmonology, Cliniques Universitaires Saint-Luc, Université Catholique de Louvain, 1200 Bruxelles, Belgium; 5Institut de Recherche Expérimentale et Clinique (IREC), Université Catholique de Louvain, 1200 Bruxelles, Belgium; 6Department of Radiology, Cliniques Universitaires Saint-Luc, Université Catholique de Louvain, 1200 Bruxelles, Belgium; 7Department of Respiratory Medicine, University Hospital (CHU) of Liège, 4000 Liège, Belgium; 8GIGA-CRC In Vivo Imaging, University of Liège, 4000 Liège, Belgium

**Keywords:** NSCLC, immune checkpoint inhibitors, radiomics, delta-radiomics, computed tomography

## Abstract

**Simple Summary:**

Accurate and early selection of patients with advanced non-small-cell lung cancer (NSCLC) who would benefit from immunotherapy is of the utmost clinical importance. The aim of our retrospective multi-centric study was to determine the potential role of CT-based radiomics machine learning models in predicting treatment response and survival in patients with advanced NSCLC treated with immune checkpoint inhibitors. Our delta-radiomics signature was able to identify patients who presented a clinical benefit at 6 months early, with an AUC obtained on an external test dataset of 0.8 (95% CI: 0.65−0.95).

**Abstract:**

The aim of our study was to determine the potential role of CT-based radiomics in predicting treatment response and survival in patients with advanced NSCLC treated with immune checkpoint inhibitors. We retrospectively included 188 patients with NSCLC treated with PD-1/PD-L1 inhibitors from two independent centers. Radiomics analysis was performed on pre-treatment contrast-enhanced CT. A delta-radiomics analysis was also conducted on a subset of 160 patients who underwent a follow-up contrast-enhanced CT after 2 to 4 treatment cycles. Linear and random forest (RF) models were tested to predict response at 6 months and overall survival. Models based on clinical parameters only and combined clinical and radiomics models were also tested and compared to the radiomics and delta-radiomics models. The RF delta-radiomics model showed the best performance for response prediction with an AUC of 0.8 (95% CI: 0.65−0.95) on the external test dataset. The Cox regression delta-radiomics model was the most accurate at predicting survival with a concordance index of 0.68 (95% CI: 0.56−0.80) (*p* = 0.02). The baseline CT radiomics signatures did not show any significant results for treatment response prediction or survival. In conclusion, our results demonstrated the ability of a CT-based delta-radiomics signature to identify early on patients with NSCLC who were more likely to benefit from immunotherapy.

## 1. Introduction 

In the last decade, immune checkpoint inhibitors (ICIs) targeting programmed death-1 (PD-1) or programmed death ligand-1 (PD-L1) have demonstrated unprecedented efficacy in increasing clinical response and survival of patients with treatment-naïve or previously treated advanced non-small cell lung cancer (NSCLC) [1,2,3,4]. The clinical benefit from immunotherapy has been correlated with PD-L1 expression on tumor cells as measured by immunohistochemistry, showing better outcomes in tumors with PD-L1 expression greater than 50% [2,5]. Tumor cell PD-L1 expression remains the cornerstone of the NSCLC treatment algorithm and the only immune-related predictive biomarker recommended by the European Society for Medical Oncology in the selection of patients who would benefit from PD-1/PD-L1 inhibitors [6]. However, PD-L1 expression is far from being a perfect biomarker; many challenges persist concerning its use as a reliable predictive biomarker, such as spatial and temporal expression heterogeneities [7,8]. Furthermore, only a subset of high PD-L1 patients will respond to treatment, and, on the other hand, clinical benefit has also been observed in patients with PD-L1-negative tumors [9,10]. Other biomarkers have proven effectiveness in predicting ICIs’ efficacy, such as the tumor mutational burden, but are not yet ready to be routinely implemented in NSCLC workup [6,11]. 

Therefore, there is a real need to look beyond PD-L1 expression and to develop new biomarkers, not to replace PD-L1, but to be associated with it in an integrated immune-related signature including different parameters which could eventually improve patient selection and enhance treatment effect. Towards this goal, radiomics could have a strong card to play. 

Radiomics is a quantitative imaging process which aims to extract high quantities of digital information from standard-of-care imaging and thereby build artificial-intelligence-based models assessing clinical outcomes such as prognostic or prediction of response to treatment [12,13]. Radiomics has numerous advantages over tissue-based biomarkers, including being non-invasive, being repeatable over time, and being able to explore the entirety of the primary tumor as well as all secondary lesions included in the imaging scan. Radiomics has shown its ability to assess the biological features of tumoral and peritumoral tissues, including local immune characteristics, such as evaluating the presence of tumor-infiltrating immune cells (CD8+, Th1 or Th2 cells) or PD-L1 expression, suggesting its capacity to play a role in the selection of patients who are more likely to respond to immunotherapy [14,15,16]. Several studies have started to explore the potential of radiomics to predict response in advanced NSCLC treated with ICIs showing variable results [17,18,19,20,21]. 

More recently, a subspecialty of radiomics called delta-radiomics has gained increasing interest [22]. Delta-radiomics consists of extracting radiomics features from the same region of interest of the same patient but at different time intervals in order to study the variations of the radiomics features over time. Compared to standard radiomics, which reflect a static situation, delta-radiomics biomarkers study the phenotypical modifications of a tissue or a lesion that may occur after the introduction of a treatment, for example. This approach may allow the detection of early signs of tumor response before any size modification, [23]. Moreover, delta-radiomics parameters seem to be more robust compared to standard radiomics features [24,25]. This approach seems particularly adapted to immunotherapy, in which the kinetics of tumor size changes are not always related to treatment response [26]. To date, only a few studies with various methodologies have investigated delta-radiomics in the setting of immunotherapy, obtaining moderate-to-good results [19,20,21,22]. However, more data are needed to move towards a potential implementation of those image-based models in routine clinical care, exploring the interest of multiple-lesion analysis and trying to embed clinical factors to delta-radiomics features to build integrated models.

Therefore, in this study, we conducted a retrospective multi-centric analysis to help determine the potential role of CT-based radiomics and delta-radiomics in predicting tumoral response and survival in patients with advanced NSCLC treated with PD-1/PD-L1 inhibitors, exploring the added value of multi-lesion analysis and clinical combined analyses.

## 2. Materials and Methods

### 2.1. Patients

We retrospectively reviewed the medical and radiological records of all patients with advanced or recurrent NSCLC treated with PD-1/PD-L1 inhibitors as monotherapy between 2015 and 2020 at the University Hospital of Liège, Belgium. The inclusion criteria were: (a) available baseline contrast-enhanced CT acquired up to two months before anti-PD-1/PD-L1 initiation; (b) presence of at least one measurable tumoral lesion according to the Response Evaluation Criteria in Solid Tumors (RECIST) version 1.1 on the baseline CT; (c) presence of at least one follow-up CT after 2 to 4 cycles of immunotherapy to assess response. The exclusion criteria were: (a) baseline CT with slice thickness greater than 3mm; (b) patients presenting with ill-defined lesions only or poor CT quality; (c) patients who received intercurrent radiotherapy or surgery on target lesions; (d) non-progressive patients with follow-up time less than 6 months. This cohort of patients constituted the training dataset.

A second dataset (external test set) of patients with advanced or recurrent NSCLC treated with anti-PD-1/PD-L1 monotherapy between 2015 and 2018 at the University Hospital of Saint-Luc, Brussels, Belgium was selected according to identical criteria. A subset of patients who had a contrast-enhanced CT at first follow-up after 2 to 4 cycles of therapy were selected for delta-radiomics analysis in each dataset. Each patient received either anti PD-1 (nivolumab 3 mg/kg every 2 weeks or pembrolizumab 200 mg every 3 weeks) or anti PD-L1 (atezolizumab 1200 mg every 3 weeks) as first-line treatment when PD-L1 expression was ≥50% (from March 2017) or as further lines regardless of PD-L1 expression. 

The following demographic and clinical data were collected: age, sex, smoking history, Eastern Cooperative Oncology Group performance status (ECOG PS), histologic subtype of NSCLC, clinical stage, ICI molecule and line, tumoral PD-L1 expression assessed via immunohistochemistry, and the presence of an oncogenic driver mutation in epidermal growth factor receptor (EGFR), anaplastic lymphoma kinase (ALK), and ROS1 when available. Pre-treatment peripheral blood biomarkers, such as absolute neutrophil count (ANC), absolute lymphocyte count (ALC), absolute eosinophil count (AEC), and neutrophil to lymphocyte ratio (NLR), were also studied.

The clinical endpoints to be predicted by the models were the treatment response status at 6 months and the overall survival (OS). Treatment response was assessed via CT or MRI every 2 to 6 cycles, according to RECIST version 1.1. Patients who presented a complete response, partial response, or stable disease at 6 months were considered responders. Patients who presented with progressive disease within 6 months after PD-1/PD-L1 inhibitor initiation were considered non-responders. OS was measured from the time of ICI initiation to death from any cause. Patients without documented clinical or radiological disease progression and who were still alive were censored at the date of last follow-up. The end of the follow-up period was 1 October 2021.

This study received approval from both institutional review boards, and the need for informed consent was waived based on its retrospective design. 

### 2.2. CT Images Acquisition and Segmentation

The pre-treatment and follow-up CT images were acquired with scanners manufactured by Siemens Healthineers (Erlangen, Germany; Somatom Emotion, Sensation, Definition, Definition AS or Edge plus), Phillips Healthcare (Amsterdam, The Netherlands; Brilliance), or GE Healthcare (Chicago, IL, USA; Brightspeed or Revolution CT) using tube voltage ranging from 80 kVp to 130 kVp, automatic adjustment of tube current, and a 512 × 512 matrix. The CT images were performed by covering either the chest or the chest and the abdomen after intravenous injection of 70 to 90 mL of iodine-based contrast. Images of the chest were obtained at the arterial phase and images of the abdomen at the portal venous phase. All images were reconstructed using a standard convolution kernel with slice thickness comprised between 0.6 and 3 mm. All CT images were independently reviewed by two radiologists (10 and 4 years of experience) to identify target lesions in the thorax and the abdomen and assess response at each time point according to RECIST 1.1. For each patient, the most substantial target lesion, called the index lesion, was also identified. All discrepancies between the two readers were reviewed until a consensus was reached. All target lesions according to RECIST 1.1 were secondarily segmented by one radiologist with 10 years of experience using semi-automatic tools with the ITK-SNAP v.3.8.0 software (www.itksnap.org accessed on 12 June 2019) on baseline and on first follow-up CT. Chest lesions were segmented on the arterial phase images and abdominal lesions on the portal venous phase.

### 2.3. Feature Extraction and Selection

#### Feature Selection

A total of 187 handcrafted radiomic features were extracted from all the segmented lesions both at baseline and at follow-up CT using the software developed by the company Radiomics (Oncoradiomics SA, Belgium), including shape, first-order, and texture features. The feature values were submitted to a logarithmic filter. The correlation between each radiomic and clinical features was assessed by evaluating the Pearson coefficient. If two features exhibited a coefficient higher than 0.9, the one with the lowest predictive power was discarded. A maximum relevance minimum redundancy (MRMR) feature-ranking algorithm was used to select a subset of promising predictors. The feature selection procedure was adapted to the machine learning method considered. The MRMR FCQ (F-test correlation quotient), WCQ (Wald test correlation quotient), and RFCQ (random forest correlation quotient based on GINI importance) variants were chosen for the classification of generalized linear models (GLMs), for Cox proportional-hazard regression models and for classification/survival random forest (RF) models, respectively. F-score calculated using ANOVA was used to assess features relevance for GLM. Wald test score was used to assess features relevance for the Cox proportional-hazard regression models. GINI importance index was used to assess features relevance for RF models. Delta-radiomic features were calculated by subtracting baseline radiomic features from follow-up radiomic features, to represent the variations between pre- and post-treatment CT radiomic features. 

### 2.4. Model Building

Regarding both outcomes, we first developed a clinical model based on clinical predictors only. The model was built following a forward selection approach using the AIC as the selection metric. Next, three different types of radiomics models were built: (a) a single-lesion radiomics model, incorporating the radiomic features of one lesion (the index lesion) per patient; (b) a multiple-lesion radiomics model, incorporating the radiomic features of all RECIST 1.1 target lesions per patient; and (c) a delta-radiomics model, studying the variations in radiomic features of the index lesion between baseline and first follow-up CT. Additionally, we developed an early RECIST model based on RECIST 1.1 assessment at first follow-up CT to be compared to our delta-radiomics model. Finally, we realized two combined analyses: (a) a combined single-lesion model, associating clinical parameters with baseline single-lesion radiomic features, and (b) a combined delta-radiomics model, combining clinical parameters with delta-radiomics features. These combinations were conducted by concatenating the radiomics signatures with the clinical one and retraining the combined models on the training set. The workflow of the study is shown in Figure 1. 

Two different types of machine learning analysis were carried out for each model. Concerning response prediction, GLM and RF signatures were built with the combinations of promising predictors. Regarding survival prediction, Cox proportional hazard regression and RF signatures were built. For each model, the most performant signatures were selected by using a 5-repetition 10-fold cross-validation approach on the whole training dataset based on the highest mean performance and trained again on the set. Concerning the RECIST model, GLM and Cox proportional hazard regression were used to build response and survival prediction signatures, respectively. All the models were then applied to the external test set to assess their true performances on independent data.

The radiomics quality score, as proposed by Lambin et al., has been estimated for our study at 17 points out of 36 (47%) [27].

### 2.5. Statistical Analysis

The Wald test was used for the univariate analysis of the clinical predictors. Performances of the response prediction models were evaluated using the area under receiver operating characteristic (ROC) curve (AUC) and their corresponding 95% confidence interval (CI). Performance of the survival models were evaluated using the Kaplan–Meier algorithm and a log rank test between low-score and high-score subsets obtained with a 50–50 split. All the statistical analyses were carried out with R software (version 4.1.1), using the following functions: the glm function, the ranger package for RF survival analysis, the caret package for RF classification, and the coxph function. 

## 3. Results

### 3.1. Clinical Characteristics

In the training dataset, 146 patients met the inclusion criteria, and a total of 345 target lesions were segmented on baseline CT and analyzed. A subset of 121 patients met the delta-radiomics inclusion criteria, in whom 284 lesions were additionally segmented on the 1st follow-up CT and included in the delta-radiomics model. In the test dataset, 42 patients met the inclusion criteria for baseline radiomics analyses, and 39 patients met the criteria for delta-radiomics analysis, in whom 91 and 84 lesions were segmented on baseline and 1st follow-up CTs, respectively. The anatomical location of each target and index lesions can be found in Appendix A. The patient flow diagram is shown in Figure 2.

The median time interval between ICI initiation and baseline CT acquisition was 21 days (range, 0−57 days) in the whole training dataset and 15 days (range, 1–53 days) in the whole test dataset. The median time interval between baseline and first follow-up CT acquisition was 70 days (range, 28−125 days) in the delta-radiomics training dataset and 69 days (range, 35−87 days) in the delta-radiomics test dataset. The median follow-up time was 15.1 months (range, 0.8−71.1) in the training dataset and 12 months (range, 1.2−73.7) in the test dataset.

Patients who presented with SD, PR, or CR at 6 months (responders) were 69/146 (47%) in the training dataset and 20/42 (48%) in the test dataset. Among the 146 patients of the training dataset, peripheral blood biomarkers were available in 136 patients (93%). The median ANC was 6.14 × 10^3^ cells/mm^3^ (range, 0.12−79.4), the median NLR was 1.5 × 10^3^ cells/mm^3^ (range, 0.31−11.26), the median NEC was 0.14 × 10^3^ cells/mm^3^ (range, 0−1.07), and the NLR was 4.1 (range, 0.09−94.52). In the test dataset, the median ANC was 5.56 × 10^3^ cells/mm^3^ (range, 0.6−70.29), the median NLR was 1.45 × 10^3^ cells/mm^3^ (range, 0.41−3.76), the median NEC was 0.11 × 10^3^ cells/mm^3^ (range, 0−0.84), and the NLR was 4.33 (range, 0.76−20.61). Tumoral PD-L1 expression was unknown in 35% of the patients in the whole dataset and in 37% of the patients in the delta-radiomics dataset, therefore this parameter was not included in our models. Demographic and clinical characteristics of the patients are reported in Table 1. Performances of the clinical characteristics regarding response at 6 months and OS are show in Table 2. The results were used to build the clinical models. 

The best clinical GLM signature for predicting response included five parameters (sex, clinical stage, ANC, AEC, and NLR) and achieved an AUC of 0.64 (95% CI: 0.46−0.82) on the external dataset (Figure 3). Concerning OS, the best clinical GLM signature included five parameters (sex, pathological type, ALC, ANC and NLR) and achieved a concordance of 0.51 (95% CI: 0.4−0.63) (Figure 4). The median OSs of the two groups obtained by splitting the dataset based on survival score were 14.1 months in the low-risk group and 11.6 months in the high-risk group.

### 3.2. Radiomic Features Selection

After correlation assessment between clinicopathological and radiomic features, no clinicopathological parameter was discarded. The lists of all the radiomic features selected for each radiomics model and their relevance (F-score for linear models and GINI importance index for RF models) can be found in the additional files (Appendix A). 

### 3.3. Radiomics Single-Lesion Analysis

Regarding the performance of the radiomics models on prediction of treatment response, the radiomics GLM signature of the index lesion included four features (stat_P90, IH_Median, NGLDM_DE, GLCM_ClusterShade) and yielded AUC values of 0.65 (95% CI: 0.56−0.74) and 0.6 (95% CI: 0.42−0.78) on the training and test datasets, respectively. The RF model based on 50 promising features yielded an AUC value of 0.66 (95% CI: 0.48−0.83) on the test dataset (Figure 3).

Regarding the survival outcome, the Cox signature of the index lesion included three features (IH_Mode, morph_Elongation, stat_Max) and achieved concordances of 0.57 (95% CI: 0.51−0.63) and 0.62 (95% CI: 0.5−0.73) on the training and test datasets, respectively. The median OS were 18.7 months in the low-risk score group and 9.8 months in the high-risk score group. The RF model based on 50 promising features showed a concordance of 0.4 on the test dataset (Figure 4).

### 3.4. Radiomics Multiple-Lesion Analysis

Regarding the performance of the radiomics models on prediction of treatment response, the radiomics GLM signature of the index lesion included four features (IH_MedianD, NGLDM_DV, GLMC_InfoCorr1, GLMC_InfoCorr2) and yielded AUC values of 0.63 (95% CI: 0.57−0.69) and 0.54 (95% CI: 0.41−0.66) on the training and test datasets, respectively. The RF model, based on 50 promising features, yielded an AUC value of 0.62 (95% CI: 0.5−0.74) on the test dataset.

Regarding the survival outcome, the Cox signature included two features (shape_Flatness, NGLDM_DE) and achieved concordances of 0.56 (95% CI: 0.52−0.6) and 0.5 (95% CI: 0.43−0.57) on the training and test datasets, respectively. The RF model based on 50 promising features showed a concordance of 0.52 on the test dataset.

### 3.5. Delta-Radiomics Analysis

Concerning treatment response prediction, the delta-radiomics GLM signature included three baseline features (IH_MedianD, GLMC_Entropy2, GLCM_InfoCorr1) and two delta-radiomic features (shape_MinorAxisLenght_delta, IH_MinGrad_delta) and yielded an AUC value of 0.84 (95% CI: 0.78−0.91) on the training dataset and an AUC value of 0.77 (95% CI: 0.61−0.93) on the test dataset. The RF model based on 50 promising features yielded an AUC value of 0.8 (95% CI: 0.65−0.95) on the test dataset (Figure 3).

Regarding the survival outcome, the Cox signature included two delta-radiomic features (Shape_Volume_delta, GLDZM_LILDE_delta) and achieved concordances of 0.70 (95% CI: 0.64−0.77) and 0.68 (95 % CI: 0.56−0.8) on the training and test datasets, respectively. The median OSs were 29.3 months in the low-risk score group and 10.5 months in the high-risk score group. The RF model based on 50 promising features showed a concordance of 0.62 on the test dataset (Figure 4). 

### 3.6. Early RECIST Analysis

Regarding response prediction, the GLM signature achieved an AUC of 0.66 (95% CI: 0.54−0.78) on the test dataset. Looking at survival prediction, the Cox signature yielded a concordance of 0.56 (95% CI: 0.47−0.65).

### 3.7. Combined Models

The combined models were obtained by associating the clinical signatures with the radiomics signatures. The combined single-lesion models showed AUCs of 0.69 (95% CI: 0.52−0.86) and 0.66 (95% CI: 0.48−0.84) on the external dataset in response prediction using GLM and RF models, respectively. The combined delta-radiomics models achieved AUCs of 0.78 (95% CI: 0.62−0.93) and 0.78 (95% CI: 0.62−0.94) using GLM and RF models, respectively (Figure 3). 

Concerning overall survival, the combined single-lesion Cox model showed a concordance of 0.62 (95% CI: 0.5−0.73) and the combined delta-radiomics Cox model a concordance of 0.65 (95% CI: 0.54−0.77) (Figure 4). The median OS were 22 months in the low-risk score group and 8.5 months in the high-risk score group for the combined single model and 21.9 and 6.3 months for the combined delta-radiomics model. A comparison of the performances of the different models on the external dataset is shown in Table 3.

## 4. Discussion

PD-1/PD-L1 inhibitors have reshaped therapeutic strategies in NSCLC, but efficient response prediction biomarkers are still lacking. 

Our single-time-point baseline radiomics signatures have shown moderate performances at predicting tumor response and survival, unlike other studies [17,28]. This could be explained by the heterogeneity of our population in terms of lesions’ locations. On the other hand, our delta-radiomics signatures, which considered feature variation at different acquisition times, have shown good performances in predicting response to immunotherapy and moderate performances in predicting OS. Delta-radiomics predictive performances were significantly better than clinical and baseline radiomics-only models both for treatment response and survival predictions. This finding is in line with the recent literature on immunotherapy response prediction, in which models incorporating feature variations over time have systematically outperformed models based on pre-treatment CT alone [19,21,29]. These results are comprehensible if the delta-radiomics models, as in our case, include volume-related features assessing early tumor shrinkage, which is the hallmark of tumor sensitivity to treatment. However, in immuno-oncology, atypical response patterns make volume-related assessment tools unreliable, justifying the need for alternative approaches. In this context, delta-radiomics is able to go beyond volumetric variations in addressing spatial and temporal structural heterogeneity within the tumor. Interestingly, our delta-radiomics signatures included a combination of size-based and texture-based features, looking at tumoral volume variation and tumoral heterogeneity, respectively. 

The composition of the tumor microenvironment (TME)—especially the quantity and the quality of tumor-infiltrating lymphocytes (TILs), which reflect the tumor immune phenotype—has been shown to influence response to PD-1/PD-L1 inhibitors [30]. Radiomics has previously shown its ability to depict the density of TILs in the TME using pre-treatment CT [15,20]. Beyond pre-treatment TME composition, several early modifications of the TME induced by ICIs, such as proliferation of CD4^+^ and CD8^+^ T-Cells and NK cells, have been shown in responders with melanoma and NSCLC [30,31,32]. 

Our hypothesis is that these early cellular modifications of the TME might be captured by delta-radiomics, partly explaining the superiority of such models in predicting response, particularly in the context of immuno-oncology [20]. One characteristic of our study is that we compared our delta-radiomics models to an early RECIST model, which focused only on size variation. The delta-radiomics approach showed better performance. Our results are in agreement with a recent study by Gong et al., in which delta-radiomics and RECIST models were also compared to predict survival in a cohort of patients with NSCLC treated with ICIs, even if our methodologies were somewhat different in terms of outcomes, feature selection processes, and machine learning models [21]. RECIST criteria remain the gold standard for assessing treatment response, but they fail in fully capturing tumor behavior complexity. It is the combination of information related to volume and heterogeneity changes that seems to account for the predictive capabilities of delta-radiomics in the context of NSCLC treated with immunotherapy [23]. 

Khorrami et al. were the first to report a CT-based delta-radiomics signature able to predict response and overall survival of patients with NSCLC treated with ICI. They focused only on single-lesion delta-radiomics, and their model yielded good performance, laying the foundations of the technique [20]. The present study aims at expanding the field by using a multiple approach, i.e., multiple types of models, using both single-time-point and delta-radiomics extracted from multiple lesions and integrating multiple clinical parameters. First, we compared delta-radiomics models to clinical and combined models, associating both clinical and radiomics features. Our radiomics and delta-radiomics models showed better performance than the clinical models alone. However, the inclusion of clinical parameters in the radiomics models did not improve prediction performance. Concerning single-time-point radiomics, these findings are inconsistent with previously published studies [28,33,34]. This could be explained by the heterogeneity of our population, gathering patients presenting different tumor stages and receiving PD-1/PD-L1 inhibitors as first-line or further lines of treatment. Regarding delta-radiomics, to date, only one study from—Xie et al.—has tried to incorporate clinical parameters into the delta-radiomics model in the setting of NSCLC treated with ICI [29]. Their combined signature showed good performances in predicting PFS and the combined models provided better clinical utility than delta-radiomics models within a reasonable threshold probability range. However, in addition to the fact that their population was different, including patients with combination therapies, it was also somewhat limited (n = 97), and the absence of an external dataset could have resulted in an overestimation of the performances of the model.

In our study, apart from PD-L1 expression, no clinical parameter showed good correlation with response and survival outcomes. These findings are in line with a recent meta-analysis from Yang et al. including 15829 NSCLC patients treated with ICI [35]. They found that survival benefit from ICI was not related to clinicopathological factors such as patients’ gender, age, PS, pathology, or smoking history. That the combined models did not yield better performances than radiomics and delta-radiomics models may thus not come as a complete surprise.

Another insight of our study is that we performed a baseline multiple-lesion radiomics analysis, which is in compliance with the RECIST assessment recommendations, to be compared to a more straightforward analysis using only one representative lesion per patient. Interestingly, despite the fact that a multiple-lesion assessment should better characterize the disease, the multiple lesions’ signatures did not show superior predictive performances compared to the single-lesion approach, in line with a recent paper from Liu et al. in which, of note, they only used unenhanced CT [19]. However, considerably more data are needed to fully comprehend the processes and consolidate the knowledge in order to implement radiomics-based biomarkers in daily practice.

Our study has also some limitations. First, though multi-centric, our study is retrospective and the sample size remains small. Our patient population is heterogeneous, including different tumor histologies, treatment lines, and patient profiles; a more accurate selection of a sub-population of patients might have resulted in better performances, especially in predictive models from baseline scans. Differences in acquisition and contrast injection protocols inside institutions may have affected radiomic feature extraction and analysis as well. There were also differences in terms of PD-L1 expression and treatment lines between the training set and the test set, and that could have impacted the performances of our models, particularly the delta-radiomics. On the positive side, these results are noteworthy because they support the applicability of delta-radiomics prediction models to large and diverse patient populations, highlighting the robustness and the generalizability of the methodological approach even across CT scans acquired in different centers using different scanners and protocols. PD-L1 expression, which is currently the only approved predictive biomarker for immunotherapy in NSCLC, was unknown in about a third of our population and was therefore not included in our models. Integration of PD-L1 expression as a parameter in our radiomics and delta-radiomics models might have improved their performances. However, in the recently published Checkmate 227 update reporting the 5-year clinical outcomes of the trial, comparing first-line immunotherapy combination to chemotherapy in metastatic NSCLC, immunotherapy achieved substantial long-term clinical benefit regardless of PD-L1 expression. The 5-year OS rates in PD-L1 negative (<1%) patients were 19% (nivolumab plus ipilimumab) versus 7% (chemotherapy), which might lead to revisiting the actual clinical impact of this parameter [36]. Additionally, we have considered responders as any patient who had not progressed at 6 months, but we did not differentiate the different types of response in our models. Finally, the time interval between baseline CT and the initiation of PD-1/PD-L1 inhibitors was wide; changes in tumoral tissues could have occurred in that time interval that could have modified the radiomics signature.

## 5. Conclusions

In conclusion, delta-radiomics is a non-invasive tool combining information related to changes in tumor volume and heterogeneity over time. Our results demonstrated the ability of a CT-based delta-radiomics signature to make an early identification of patients who are more likely to benefit from PD-1/PD-L1 inhibitors in the setting of advanced or recurrent NSCLC. 

## Figures and Tables

**Figure 1 cancers-15-01968-f001:**
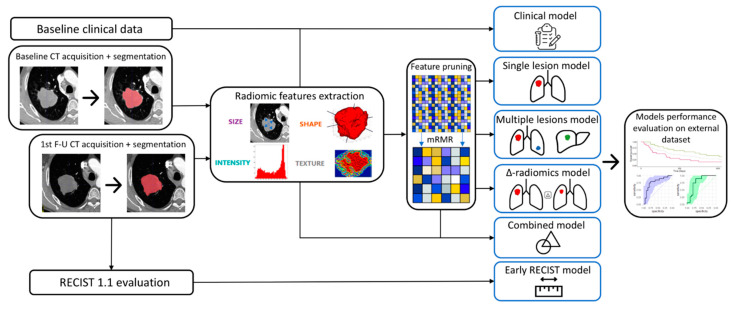
Radiomics workflow of the study.

**Figure 2 cancers-15-01968-f002:**
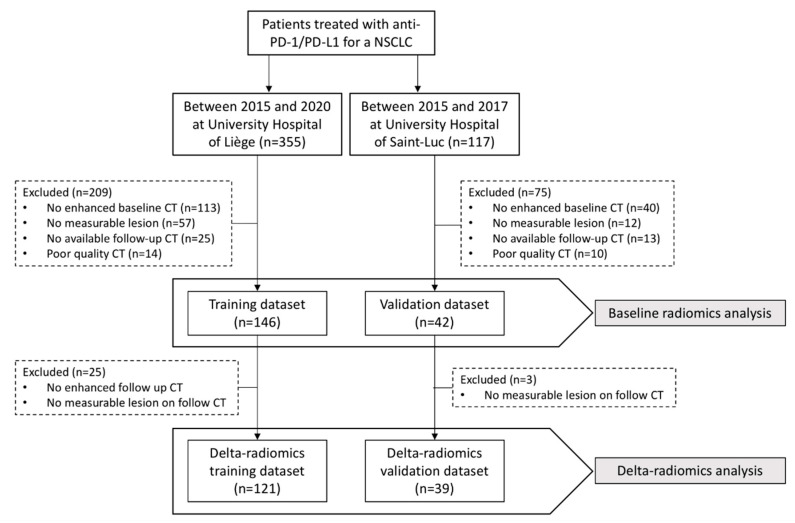
Patient flow diagram.

**Figure 3 cancers-15-01968-f003:**
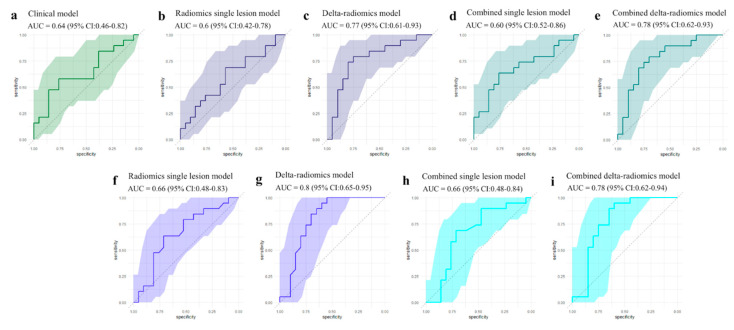
The ROC curves for treatment response prediction of the generalized linear (**a**–**e**) and the random forest (**f**–**i**) models obtained on the external dataset.

**Figure 4 cancers-15-01968-f004:**
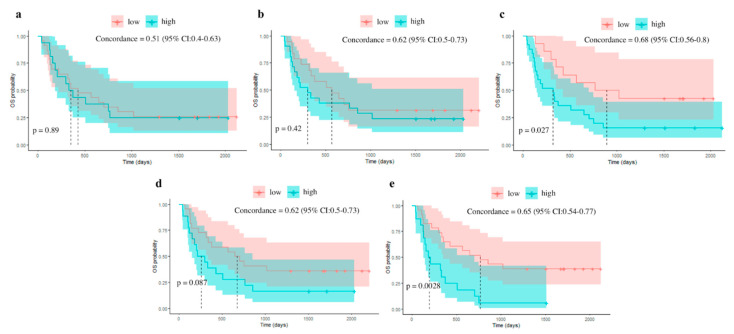
External dataset OS Kaplan–Meier curves with split based on Cox proportional hazard for the clinical model (**a**); the radiomics single-lesion model (**b**); the delta-radiomics model (**c**); the combined radiomics single-lesion model (**d**); and the combined delta-radiomics model (**e**).

**Table 1 cancers-15-01968-t001:** Demographic and clinical characteristics of patients.

Characteristics	Baseline Radiomics Analysis			Delta-Radiomics Analysis		
	All Patients (N = 188)	Training Set (N = 146)	Test Set (N = 42)	All Patients (N = 160)	Training Set (N = 121)	Test Set (N = 39)
Age, median (range)	66 (42−95)	66 (42−85)	64 (44−95)	65 (42−95)	65 (42−85)	64 (44−95)
Sex						
Male	113 (60%)	90 (62%)	23 (55%)	96 (60%)	76 (63%)	20 (51%)
Female	75 (40%)	56 (38%)	19 (45%)	64 (40%)	45 (37%)	19 (49%)
Smoking history						
Never	10 (5%)	3 (2%)	7 (17%)	9 (6%)	2 (2%)	7 (18%)
Current or former	178 (95%)	143 (98%)	35 (83%)	151 (94%)	119 (98%)	32 (82%)
Pathological type						
Adenocarcinoma	117 (62%)	86 (59%)	31 (74%)	100 (62%)	71 (59%)	29 (74%)
Squamous cell	58 (31%)	50 (34%)	8 (19%)	48 (30%)	41 (34%)	7 (18%)
Other ^a^ Clinical stage Stage III Stage IV PD-L1 expression Unknown <1% 1−49% ≥50%	13 (7%) 27 (14%) 161 (86%) 66 (35%) 30 (16%) 26 (14%) 66 (35%)	10 (7%) 26 (18%) 120 (82%) 47 (32%) 19 (13%) 20 (14%) 60 (41%)	3 (7%) 1 (2%) 41 (98%) 19 (45%) 11 (26%) 6 (14%) 6 (14%)	12 (8%) 22 (14%) 138 (86%) 59 (37%) 27 (17%) 20 (12%) 54 (34%)	9 (7%) 21 (17%) 100 (83%) 42 (35%) 17 (14%) 14 (12%) 48 (39%)	3 (8%) 1 (3%) 38 (97%) 17 (43%) 10 (26%) 6 (15%) 6 (15%)
Known mutation ^b^	6 (3%)	3 (2%)	3 (7%)	6 (4%)	3 (2%)	3 (8%)
Treatment molecule						
Pembrolizumab	67 (36%)	61 (42%)	6 (14%)	51 (32%)	46 (38%)	5 (13%)
Nivolumab	100 (53%)	71 (49%)	29 (69%)	90 (56%)	63 (52%)	27 (69%)
Atezolizumab	21 (11%)	14 (9%)	7 (17%)	19 (12%)	12 (10%)	7 (18%)
Treatment line						
First line	50 (27%)	47 (32%)	3 (7%)	38 (24%)	35 (29%)	3 (8%)
Further lines	138 (73%)	99 (68%)	39 (93%)	122 (76%)	86 (71%)	36 (92%)
Response at 6 months						
Responders	89 (47%)	69 (47%)	20 (48%)	77 (48%)	58 (48%)	19 (49%)
Non-responders	99 (53%)	77 (53%)	22 (52%)	83 (52%)	63 (52%)	20 (51%)
OS ^c^, median (range)	14.9 (0.4−73.7)	15.1 (0.4−71)	12 (1.2−73.7)	15 (1.2−70.8)	15.3 (1.6−60.4)	12.4 (1.2−70.8)

OS, overall survival; ^a^ Other includes large cell carcinoma and carcinoma not otherwise specified; ^b^ including epidermal growth factor receptor (EGFR), anaplastic lymphoma kinase (ALK), or ROS1 mutations; ^c^ in months.

**Table 2 cancers-15-01968-t002:** Univariate analysis of the clinical predictors.

Clinical Predictors	Response at 6 Months		OS	
	AUC (95% CI)	*p*-Value	C-Index (95% CI)	*p*-Value
Age	0.51 (0.41−0.51)	0.97	0.52 (0.46−0.58)	0.41
Sex	0.55 (0.47−0.55)	0.24	0.51 (0.46−0.56)	0.42
Clinical stage	0.54 (0.47−0.54)	0.26	0.52 (0.47−0.56)	0.50
Line of treatment	0.54 (0.46−0.54)	0.28	0.55 (0.50−0.60)	0.06
Pathological type	0.50 (0.42−0.50)	0.92	0.50 (0.45−0.56)	0.63
ICI molecule	0.53 (0.48−0.53)	0.21	0.52 (0.49−0.55)	0.18
Absolute neutrophil count	0.59 (0.49−0.59)	0.07	0.59 (0.52−0.66)	0.05
Absolute lymphocyte count	0.50 (0.40−0.50)	0.37	0.47 (0.41−0.53)	0.22
Absolute eosinophil count	0.60 (0.51−0.61)	0.05	0.57 (0.51−0.64)	0.13
Neutrophil to lymphocyte ratio	0.54 (0.45−0.54)	0.41	0.57 (0.51−0.64)	0.43

ICI, immune checkpoint inhibitor; AUC, area under the ROC curve; OS, overall survival.

**Table 3 cancers-15-01968-t003:** Performances of the predictive models on the external dataset.

Predictive Models	Response at 6 Months		OS	
	GLM AUC (95% CI)	RF	Cox Model C-Index (95% CI)	RF
Clinical model	0.64 (0.46−0.82)	/	0.51 (0.4−0.63)	/
Single-lesion radiomics model	0.6 (0.42−0.78)	0.66 (0.48−0.63)	0.62 (0.51−0.73)	0.4
Multiple-lesion radiomics model	0.54 (0.41−0.66)	0.62 (0.5−0.74)	0.5 (0.43−0.57)	0.52
Delta-radiomics model	0.77 (0.61−0.93)	0.8 (0.65−0.95)	0.68 (0.56−0.8)	0.62
RECIST model	0.66 (0.54−0.78)	/	0.56 (0.47−0.65)	/
Combined single-lesion model	0.69 (0.52−0.86)	0.66 (0.48−0.84)	0.62 (0.5−0.73)	/
Combined delta-radiomics model	0.78 (0.62−0.93)	0.78 (0.62−0.94)	0.65 (0.54−0.77)	/

GLM, generalized linear model; RF, random forest; AUC, area under the ROC curve; OS, overall survival.

## Data Availability

The datasets used and/or analyzed during the current study are available from the corresponding author on reasonable request.

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
