# Peer review of "Radiomics and Delta-Radiomics Signatures to Predict Response and Survival in Patients with Non-Small-Cell Lung Cancer Treated with Immune Checkpoint Inhibitors"

_cancers, 2023, doi:10.3390/cancers15071968_

Round 1

Reviewer 1 Report

1.     The introduction lacks an explanation of what delta-radiomics is, its advantages, and its latest research advances in predicting the efficacy of immunotherapy in NSCLC.

2.     The studies using radiomics or delta-radiomics to predict the efficacy of immunotherapy in NSCLC have been reported, and the authors have not shown the highlights and innovations of this study compared with other studies.

3.     The authors only described the obtained results in the abstract, but didn’t explain the clinical application values of this study.

4.     In the discussion, the authors referred to an article that also used delt-radiomics to predict the efficacy of NSCLC ICI [20], in which it was found that the AUC of the DelRADx combined classification analysis method in distinguishing between responders and non-responders was 0.88±0.08, and in the two validation sets, the AUC of the DelRADx classification method was 0.85 (n = 62) and 0.81 (n = 27), respectively. What are the innovation points and application values of this study compared with the mentioned article? The authors need to answer this question in the discussion.

5.     In the Delta-radiomics analysis, there were significant differences in clinical characteristics between the training set and test set, such as the proportion of patients with high PD-L1 expression of 39% and 15%, respectively, and the proportion of first-line therapy of 29% and 8%, respectively, which may be potential factors influencing the results.

6.     In the Delta-radiomics model, the low-score subset had a significant survival benefit compared to the high-score subset, but no specific survival time was described for both subsets to assess the extent of clinical benefit.

7.     It was mentioned in the discussion that radiomics might predict efficacy and survival by influencing the tumor microenvironment, but this was not explored in this study.

Author Response

First, we would like to thank the reviewer for his valuable comments.

You will find hereafter the comments and suggestions of the reviewers listed in bold and our specific point by point responses.

  1. The introduction lacks an explanation of what delta-radiomics is, its advantages, and its latest research advances in predicting the efficacy of immunotherapy in NSCLC.

The introduction section has been modified to provide more details about the delta-radiomics processes. The advantages of delta-radiomics over the standard radiomics analysis have also been further described, along with latest research in the setting of immunotherapy in NSCLC, and further references have been added to support the points (Nardone et al. 2019, ref. 24; Plautz et al. 2019, ref. 25; Xie et al. 2022, ref. 29).

  1. The studies using radiomics or delta-radiomics to predict the efficacy of immunotherapy in NSCLC have been reported, and the authors have not shown the highlights and innovations of this study compared with other studies.

The discussion has been revised to clarify the innovations of the present study compared to other delta-radimics studies, which can be summarized as follow:

  • The comparison between single and multiple lesions radiomics analysis using enhanced CT examinations.
  • The extended comparison between clinical, radiomics/delta-radiomics, RECIST and combined models.

While the studies available in the literature focused on one methodological aspect of the analysis, we tried to propose an extended comparison between different types of models and analysis (multi-lesions / single lesion; single-time-point / delta-radiomics; clinical / radiomics / combined models; simple / more complex machine learning analysis, response/survival outcomes) to help refine and optimise the processes, all using a consistent cohort and an external validation dataset.

  1. The authors only described the obtained results in the abstract, but didn’t explain the clinical application values of this study.

We have added a sentence to that effect at the end of the abstract section.

  1. In the discussion, the authors referred to an article that also used delta-radiomics to predict the efficacy of NSCLC ICI [20], in which it was found that the AUC of the DelRADx combined classification analysis method in distinguishing between responders and non-responders was 0.88±0.08, and in the two validation sets, the AUC of the DelRADx classification method was 0.85 (n = 62) and 0.81 (n = 27), respectively. What are the innovation points and application values of this study compared with the mentioned article? The authors need to answer this question in the discussion.

Khorrami et al. is obviously a key reference, as they were the first to develop a delta-radiomics model to predict tumoral response and survival in a cohort of 139 NSCLC patients. They also evaluated the correlation between delta-radiomics features with tumor-infiltrating lymphocytes on biopsies for 36 patients. The innovative aspects of our study have been further described in the discussion. This issue has also been partially addressed in the point 2 of the present review.

  1. In the Delta-radiomics analysis, there were significant differences in clinical characteristics between the training set and test set, such as the proportion of patients with high PD-L1 expression of 39% and 15%, respectively, and the proportion of first-line therapy of 29% and 8%, respectively, which may be potential factors influencing the results.

Models were trained on the population from the University hospital of Liège and then tested on the external independent population from the University hospital of Saint-Luc Brussels to evaluate their performance. Indeed, the external population presents some differences in term of clinical and pathological parameters, more particularly in terms of PD-L1 expression and treatment lines as correctly pointed out by the Reviewer. Treatment line and PD-L1 expression are linked because only patients with high PD-L1 TPS score are eligible for first-line immunotherapy. Those differences could explain why some models performed moderately well. However, the fact that some of the models did perform reasonably well, especially the delta-radiomics models, adds to the robustness and generalizability of the results, which is, in our view, a strength of the study. This potential issue has nonetheless been added to the limitations of the study.

  1. In the Delta-radiomics model, the low-score subset had a significant survival benefit compared to the high-score subset, but no specific survival time was described for both subsets to assess the extent of clinical benefit.

Median survival times for each high and low-risk group were added to the result section to illustrate the magnitude of clinical benefit of the different models.

  1. It was mentioned in the discussion that radiomics might predict efficacy and survival by influencing the tumor microenvironment, but this was not explored in this study.

Indeed, we did not explore the relationship between the radiomics signatures and the composition of the tumoral micro-environment. We have rephrased the sentence in order to clearly state that it remains a hypothesis.

Reviewer 2 Report

This paper represents an outstanding paper able to identify the role of radiomics in the prediction of clinical benefit in NSCLC patients under ICIs. In my opinion, some minor considerations should be impelemented to accept this paper for the publication

- In  the study design section, pelase, could the authors overview pathological and, when available, moelcular features of enrolled cases' In my opinion this point may improve the readibility of the manuscript.

- In the methodological section, please, could the authors explain if a tarining cohort may be usefull to improve the quality of data from patient's population?

- In the results section, could ICIs drug may represent a criteria of stratification? Could the authors discuss this point?

- In the methodological section, please, coudl the authors report criteria for positive selection to ICIs? In my opinion, the authors should explain in details the inclusion criteria related to PD.L1 expression in order to consider if borderline results may represent an outlier in data analysis

Author Response

First, we would like to thank the reviewer for his valuable comments.

You will find hereafter the comments and suggestions of the reviewer listed in bold and our specific point by point responses.

This paper represents an outstanding paper able to identify the role of radiomics in the prediction of clinical benefit in NSCLC patients under ICIs. In my opinion, some minor considerations should be impelemented to accept this paper for the publication.

1. In the study design section, pelase, could the authors overview pathological and, when available, moelcular features of enrolled cases' In my opinion this point may improve the readibility of the manuscript.

We have added this information in the result section and completed the Table 1.

2. In the methodological section, please, could the authors explain if a tarining cohort may be usefull to improve the quality of data from patient's population?

We used the cohort of 188 patients from the first hospital (University of Liège) as the training cohort to build and train the models using cross-validation method. We secondarily tested the models on the 46 patients from the external dataset (University of Saint-Luc Brussels) to better estimate the true performances of our models. Changes have been made to the methodological section for greater clarity concerning this process.

3. In the results section, could ICIs drug may represent a criteria of stratification? Could the authors discuss this point?

The type of ICI molecule (anti PD-1/ anti PD-L1) was studied as an independent clinical parameter regarding response at 6 month and OS and used in the combined models. The predictive value of ICI molecule was not significant (AUC of 0.53 p=0.21 for response prediction and C-index of 0.52 p=0.18 concerning survival). This data is reported in Table 2.

4. In the methodological section, please, could the authors report criteria for positive selection to ICIs? In my opinion, the authors should explain in details the inclusion criteria related to PD.L1 expression in order to consider if borderline results may represent an outlier in data analysis.

We have clarified the criteria for immunotherapy eligibility regarding PD-L1 expression in the methodological section as suggested.

Reviewer 3 Report

In the era of immune checkpoint inhibitors (ICIs), the identification of prognostic/predictive biomarkers appears mandatory. The development of non-invasive computer-assisted prognostic tools, specifically relying on radiomics and machine learning techniques, able to aid patient selection for ICI administration, might significantly and positively impact on clinical management and therapeutic strategies in advanced NSCLC patients.  From this perspective the manuscript by Cousin et al. addresses a clinically relevant topic. Technically and methodologically the work appears robust and the sample size, in terms of training/validation cohorts, is adequate, although not impressive. The results are well-written and followed a logical sequence. Thus, I consider the present study as meritorious of future publication in this journal, after the revisions reported below.

-        The authors have focused on radiomic and clinical parameters, identifying potential independent prognostic factors; nonetheless, they have not mentioned the only approved biomarker of ICI response: PD-L1 status (likely due to the unknown status in the majority of cases). Please, expand and cover this issue, also reporting the impact or lack of impact of PD-L1 status on ICI response and potential correlations with other clinical and radiomic features features.

-        A second point that needs to be expanded regards the potential correlation between radiomic and clinico-pathological characteristics. Did the authors explore this relevant issue? If yes, I suggest to report the data; if no, I suggest to perform this specific analysis.

-        In the Discussion section, the authors pointed out the relevance of a delta-radiomic approach, but without expanding enough the part related to clinical features and its integration in a combined radio-immune model. I suggest implementing the Discussion also reporting and commenting on other works which have employed similar integrated radiomic-clinical approaches.

Author Response

First, we would like to thank the reviewer for his valuable comments.

You will find hereafter the comments and suggestions of the reviewer listed in bold and our specific point by point responses.

In the era of immune checkpoint inhibitors (ICIs), the identification of prognostic/predictive biomarkers appears mandatory. The development of non-invasive computer-assisted prognostic tools, specifically relying on radiomics and machine learning techniques, able to aid patient selection for ICI administration, might significantly and positively impact on clinical management and therapeutic strategies in advanced NSCLC patients.  From this perspective the manuscript by Cousin et al. addresses a clinically relevant topic. Technically and methodologically the work appears robust and the sample size, in terms of training/validation cohorts, is adequate, although not impressive. The results are well-written and followed a logical sequence. Thus, I consider the present study as meritorious of future publication in this journal, after the revisions reported below.

1. The authors have focused on radiomic and clinical parameters, identifying potential independent prognostic factors; nonetheless, they have not mentioned the only approved biomarker of ICI response: PD-L1 status (likely due to the unknown status in the majority of cases). Please, expand and cover this issue, also reporting the impact or lack of impact of PD-L1 status on ICI response and potential correlations with other clinical and radiomic features features.

First line ICI are reimbursed in Belgium since March 2017 for patients with PD-L1 expression ≥ 50%. Therefore, PD-L1 expression was not systematically assessed in patients treated before March 2017. Because we made the choice to integrate data from patients treated before March 2017 in our analysis to perform a more powerful radiomics analysis, the PD-L1 status was unknown in 35% of the patients in the whole dataset and 37% of patients in the delta-radiomics dataset, and this is why we chose to exclude this parameter from the analysis. It is true that this information was not clearly discussed. We have clarified this issue in the limitation of the study, we have also added a reference (JR Brahmer et al. Journal of Clinical Oncology, 2023, ref. 36) to better cover the issue. We have also given more details about the inclusion criteria related to PD-L1 expression in the methodological section for greater clarity.

2. A second point that needs to be expanded regards the potential correlation between radiomic and clinico-pathological characteristics. Did the authors explore this relevant issue? If yes, I suggest to report the data; if no, I suggest to perform this specific analysis.

The correlation between radiomic and clinicopathological features was explored during the feature selection step which preceded the development of the models. The correlation between each radiomic and clinical features was assessed by evaluating the Pearson coefficient. If two features exhibited a coefficient higher than 0,9 the one with the lowest predictive power was discarded. And no clinicopathological characteristics was excluded from the analysis. We have added that information in the methodological and results sections to address this relevant question.

3. In the Discussion section, the authors pointed out the relevance of a delta-radiomic approach, but without expanding enough the part related to clinical features and its integration in a combined radio-immune model. I suggest implementing the Discussion also reporting and commenting on other works which have employed similar integrated radiomic-clinical approaches.

We have further developed the part related to the clinical factors, their integration in our models and discussed other studies. We have also added two references (Yang et al. 2022, ref. 35 ; Xie et al. 2022, ref. 36) to enrich the discussion.